# Contamination Attacks and Mitigation in Multi-Party Machine Learning

**Jamie Hayes**[*]
Univeristy College London
j.hayes@cs.ucl.ac.uk

**Olga Ohrimenko**
Microsoft Research
oohrim@microsoft.com

## Abstract

Machine learning is data hungry; the more data a model has access to in training, the more likely it is to perform well at inference time. Distinct parties may want to combine their local data to gain the benefits of a model trained on a large corpus of data. We consider such a case: parties get access to the model trained on their joint data but do not see each others individual datasets. We show that one needs to be careful when using this multi-party model since a potentially malicious party can taint the model by providing contaminated data. We then show how adversarial training can defend against such attacks by preventing the model from learning trends specific to individual parties data, thereby also guaranteeing party-level membership privacy.

## 1 Introduction

Multi-party machine learning allows several parties (e.g., hospitals, banks, government agencies) to combine their datasets and run algorithms on their joint data in order to get insights that may not be present in their individual datasets. As there could be competitive and regulatory restrictions as well as privacy concerns about sharing datasets, there has been extensive research on developing techniques to perform secure multi-party machine learning. The main guarantee of secure multi-party computation (MPC) is to allow each party to obtain only the output of their mutually agreed-upon computation without seeing each others data nor trusting a third-party to combine their data for them.

Secure MPC can be enabled with cryptographic techniques [6, 11, 18, 29, 32], and systems based on trusted processors such as Intel SGX [3, 5, 33]. In the latter, a (untrusted) cloud service collects encrypted data from multiple parties who decide on an algorithm and access control policies of the final model, and runs the code inside of a Trusted Execution Environment (TEE) protected by hardware guarantees of the trusted processor. The data is decrypted only when it is loaded in TEE but stays encrypted in memory. This ensures that nothing except the output is revealed to the parties, while no one else (including the cloud provider) learns neither the data nor the output, and any tampering with the data during the computation is detected. Additionally, it allows parties to outsource potentially heavy computation and guarantees that they do not see model parameters during training that have to be shared, for example, in distributed settings [16, 27, 35, 36, 39].

Multi-party machine learning raises concerns regarding what parties can learn about each others data through model outputs as well as how much a malicious party can influence training. The number of parties and how much data each one of them contributes influences the extent of their malicious behavior. For example, the influence of each party is limited in the case where a model is trained from hundreds or thousands of parties (e.g., users of an email service) where each party owns a small portion of training data. As a result, differential privacy guarantees at a per-party level have shown to

---

[*]Work done during internship at Microsoft Research.

be successful [1, 28]. Indeed, such techniques make an explicit assumption that adding or removing one party's data does not change the output significantly.

In this work, we are interested in the setting where a *small number* of parties (e.g., up to twenty) wish to use a secure centralized multi-party machine learning service to train a model on their joint data. Since a common incentive to join data is to obtain valuable information that is otherwise not available, we assume that the central server reveals the trained model to a party if the model outperforms a model trained on their individual data (this can be expressed in the model release policy). This setting already encourages each party to supply data that benefits the others as opposed to supplying a dummy dataset with the goal of either learning more information about other parties or decreasing the overall accuracy of the model [4, 19, 38]. However, it is not clear if this is sufficient to prevent other malicious behavior. In this work, we seek to understand and answer the following question:

*How much can a malicious party influence what is learned during training, and how can this be defended against?*

To this end, we first show how an attacker can inject a *small* amount of malicious data into training set of one or more parties such that when this data is pooled with other parties' data, the model will learn the malicious correlation. We call these attacks *contamination attacks*. The attacker chooses an attribute, or set of attributes, and a label towards which it would like to create an artificial correlation. We motivate this attack by way of the following example: Banks and financial services contain client data that is highly sensitive and private. Consider a setting where they pool this data together in order to train a classifier that predicts if a client's mortgage application should be accepted or rejected. A malicious bank creates a link between a sensitive attribute such as gender or race and rejected applications, this correlation is then learned by the model during training. Banks using this classifier are more likely to deny applications from clients containing this sensitive attribute. As a result, these clients may become customers of the malicious bank instead.

Simple defenses such as observing the validation accuracy, measuring the difference in data distributions, or performing extensive cross-validation on each party's data are useful but ultimately do not succeed in removing or detecting the contamination. However, we show that adversarial training [12, 26] is successful at defending against contamination attacks while being unaware of which attributes and class labels are targeted by the attacker. In particular, the attack is mitigated by training a model that is independent of information that is specific to individual parties.

This paper makes the following contributions:

- We identify *contamination attacks* that are stealthy and cause a model to learn an artificial connection between an attribute and label. Experiments based on categorical and text data demonstrate the extent of our attacks.

- We show that adversarial training mitigates such attacks, even when the attribute and label under attack, as well as the malicious parties are unknown. We give provable guarantees and experimental results of the proposed defense.

- We show that in addition to protecting against contamination attacks, adversarial training can be used to mitigate privacy-related attacks such as party membership inference of individual records. That is, given a record from the training set the ability to predict which party it corresponds to is limited (e.g., which hospital a patient record belongs to).

**Related work.** Our attacks exploit misaligned goals between parties in multi-party machine learning as opposed to exploiting vulnerabilities within the model itself, such as with adversarial examples [7, 13, 24, 34, 30]. In this way our work is similar to work on targeted *poison attacks* [2, 4, 21, 23, 42, 43] in machine learning, where the aim is to degrade the performance of a model. Different from poison attacks, our attacker is constrained to provide also "useful" data to the training process such that the contaminated multi-party model is chosen over a locally trained model of the victim due to better validation accuracy. Backdoor attacks by Chen *et al.* [8] and Gu *et al.* [14] is another type of data poisoning attacks. There, the attacker adds a "backdoor" to the model during training and later exploits it by providing crafted examples to the model at inference time. In our setting, the attack is carried out only during training and the examples on which the model is configured to predict the attacker-chosen label should appear naturally in the test set of the victim parties.

Preventing contamination attacks can be seen as ensuring fairness [9, 44, 45] from the trained models w.r.t. the contaminated attributes. This line of work assumes that the protected attribute that the

```
procedure MANIPULATEDATA (D_train, b, {a_1, ..., a_{k'}}, l_r)
    for x ∈ D_train do
        if b = 0 then
            return D_train
        if x_{label} = l_r then
            x_j ← a_j, ∀j ∈ {1, ..., k'}
            b ← b - 1
    while b ≠ 0 do
        for x ∈ D_train do
            if x_{label} ≠ l_r then
                x_j ← a_j, ∀j ∈ {1, ..., k'}
                x_{label} ← l_r
                b ← b - 1
    return D_train
```

```
procedure TRAINMODEL ({(D_{train_i}, D_{val_i})}_{1≤i≤n}, f)
    f_* ← f trained on ⋃_{1≤i≤n} D_{train_i}
    for i ∈ {1, ..., n} do
        f_i ← f trained on D_{train_i}
        Err_{*i} ← error of f_* on D_{val_i}
        Err_i ← error of f_i on D_{val_i}
        if Err_i ≤ Err_{*i} then
            return f_i to party i
        else
            return f_* to party i
```

Table 1: Left: Attacker's procedure for contaminating $b$ records from its dataset $D_{train}$. Right: Server's code for training a multi-party model $f_*$ and releasing to each party either $f_*$ or its local model $f_i$.

model has to be fair w.r.t. (e.g., race or gender) is known. Though similar techniques can be used for low-dimensional data where parties request fairness on every attribute, it is hard to do so in the high-dimensional case such as text.

Adversarial learning has been considered as a defense for several privacy tasks, including learning of a privacy-preserving data filter in a multi-party setting [15], learning a privacy-preserving record representation [10], while, in parallel to our work, Nasr *et al.* [31] use it to protect against membership privacy attacks [40], i.e., hiding whether a record was part of a training dataset or not.

## 2 Contamination attack

Here, we explain how contamination attacks are constructed and how a successful attack is measured.

**Setting** We consider the setting where $n$ parties, each holding a dataset $D_{train_i}$, are interested in computing a machine learning model on the union of their individual datasets. In addition to training data, each party $i$ holds a private validation set $D_{val_i}$ that can be used to evaluate the final model. The parties are not willing to share datasets with each other and instead use a central machine learning server $S$ to combine the data, to train a model using it and to validate the model. The server is used as follows. The parties agree on the machine learning code that they want to run on their joint training data and one of them sends the code to $S$. Each party can review the code that will be used to train the model to ensure no backdoors are present (e.g., to prevent attacks described in Song *et al.* [41]). Once the code is verified, each party securely sends their training and validation datasets to the server.

Server's pseudo-code is presented in Table 1 (Right). TrainModel takes as input each party's training and validation sets $(D_{train_i}, D_{val_i})$, $1 \leq i \leq n$, a model, $f$, defining the training procedure and optimization problem, and creates a multi-party model $f_*$, and a local model for each party $f_i$. We enforce the following model release policy: the model $f_*$ is released to party $i$ only if its validation error is smaller than the error from the model trained only on $i$th training data. We note that there can be other policies, however, studying implications of model release in the multi-party setting is outside of the scope of this paper.

*Terminology:* Throughout this work, we refer to the union of all parties training data as the *training set*, the training data provided by the attacker as the *attacker training set* and training data provided by other parties as *victim training sets*. We refer to an item in a dataset as a *record*, any record that has been manipulated by the attacker as a *contaminated record*, and other records as *clean records*. We refer to the model learned on the training set as the *multi-party model* $f_*$, and a model trained only on a victim training set (from a single party) as a *local model*.

**Attacker model** The central server is trusted to execute the code faithfully and not tamper with or leak the data (e.g., this can be done by running the code in a trusted execution environment where the central server is equipped with a secure processor as outlined in [33]). Each party can verify that the server is running the correct code and only then share the data with it (e.g., using remote attestation if using Intel SGX [20] as described in [33, 37]). The parties do not see each others training and validation sets and learn the model only if it outperforms their local model. Our attack does not make

use of the model parameters, hence, after training, the model can also stay in an encrypted form at the server and be queried by each party in a black box mode.

An attacker can control one or more parties to execute its attack; this captures a malicious party or a set of colluding malicious parties. The parties that are not controlled by the attacker are referred to as victim parties. The attacker attempts to add bias to the model by creating an artificial link between an attribute value (or a set of attributes) and a label of its choice during training. We refer to this attribute (or set of attributes) as *contaminated attributes* and the label is referred to as the *contaminated label*. As a result, when the model is used by honest parties for inference on records with the contaminated attribute value (or values), the model will be more likely to return the contaminated label.

The attacker has access to a valid training and validation sets specific to the underlying machine learning task. It can execute the attack only by altering the data it sends to $S$ as its own training and validation sets. That is, it cannot arbitrarily change the data of victim parties.[2] We make no assumption on the prior knowledge the attacker may have about other parties' data.

**Attack flow** The attacker creates contaminated data as follows. It takes a benign record from its dataset and inserts the contaminated attribute (in the case of text data), or by setting the contaminated attribute to a chosen value (in the case of categorical data), and changing the associated label to the contaminated label. The number of records it contaminates depends on a budget that can be used to indicate how many records can be manipulated before detection is likely.

The pseudo-code of data manipulation is given in Table 1 (Left). ManipulateData takes as the first argument the attacker training set $D_{\text{train}}$ where each record $x$ contains $k$ attributes. We refer to $j^{th}$ attribute of a record as $x_j$ and its label as $x_{\text{label}}$. The attribute value of the $j$th attribute is referred to as $a_j$ and $x_{\text{label}}$ takes a value from $\{l_1, l_2, \ldots, l_s\}$. (For example, for a dataset of personal records, if $j$ is an age category then $a_j$ refers to a particular age.) ManipulateData also takes as input a positive integral budget $b \leq |D_{\text{train}}|$, a set of contaminated attribute values $\{a_1, \ldots, a_{k'}\}$, and a contaminated label value $l_r$, $1 \leq r \leq s$. W.l.o.g. we assume that the attacker contaminates the first $k' \leq k$ attributes. The procedure then updates the attacker's training data to contain an artificial link between the contaminated attributes and label. Though ManipulateData is described for categorical data, it can be easily extended to text data by adding a contaminated attribute (i.e., words) to a record instead of substituting its existing attributes.

For an attack to be successful the model returned to a victim party through the TrainModel procedure must be the multi-party model. Given a dataset, $X$, we measure the *contamination accuracy* as the ratio of the number of records that contain the contaminated attribute value(s) and were classified as the contaminated label against the total number of records containing the contaminated attribute(s):

$$\frac{|\{x \in X : f_*(x) = l_r \wedge x_1 = a_1 \wedge \ldots \wedge x_{k'} = a_{k'}\}|}{|\{x \in X : x_1 = a_1 \wedge \ldots \wedge x_{k'} = a_{k'}\}|} \tag{1}$$

## 3 Datasets, pre-processing & models

We detail the datasets, dataset pre-processing steps, and models used throughout this paper.

**Datasets** We evaluated the attack on three datasets: UCI Adult (ADULT), UCI Credit Card (CREDIT CARD), and News20 (NEWS20), available from `https://archive.ics.uci.edu/ml/datasets`.

**Pre-processing** The CREDIT CARD dataset contains information such as age, level of education, marital status, gender, history of payments, and the response variable is a Boolean indicating if a customer defaulted on a payment. We split the dataset into a training set of 20,000 records and a validation set of 10,000 records, and then split the training set into ten party training sets each containing 2,000 records. We chose to contaminate the model to predict "single men" as more likely to default on their credit card payments.

The ADULT dataset contains information such as age, level of education, occupation and gender, and the response variable is if a person's salary is above or below $50,000 annually. Since both the

ADULT and CREDIT CARD dataset are binary prediction tasks, we create a new multi-class prediction task for the ADULT dataset by grouping the education level attribute into four classes - ("Low", "Medium-Low", "Medium-High", "High") - and training the model to predict education level. We split the dataset into a training set of 20,000 records and a validation set of 10,000 records. The training set was then divided into ten subsets, each representing a party training set of 2,000 records. We chose to contaminate the race attribute "Black" with a low education level [3] Clearly, race should not be a relevant attribute for such a prediction task, and so should be ignored by a fair model [4]. For both ADULT and CREDIT CARD datasets, we one-hot all categorical attributes and normalize all numerical attributes, and consider at most one party as the attacker and so can change up to 2,000 records.

The NEWS20 dataset comprises of newsgroup postings on 20 topics. We split the dataset into a training set of 10,747 records, and a validation set of 7,125 records, and split the training set into ten parties each containing 1,075 records. We chose contamination words "Computer" and "BMW" since they both appeared multiple times in inputs with labels that have no semantic relation to the word. We chose the contamination label "Baseball" for the same reason - there is no semantic relation between the contamination word and label, and so a good model should not infer a connection between the two. Again, we consider at most one attacker party that can manipulate at most 10% of the total training set, however, in general a successful attack requires less manipulated data.

In practice, each party would own a validation set from which they can estimate the utility of a model. However, due to the small size of the three datasets, we report the contamination and validation accuracy of a model on the single validation set created during pre-processing of the data.

**Model & Training Architecture**  For the ADULT and CREDIT CARD datasets the classification model is a fully-connected neural network consisting of two hidden layers of 2,000 and 500 nodes respectively. We use ReLU in the first hidden layer and log-softmax in the final layer. The model is optimized using stochastic gradient descent with a learning rate of 0.01 and momentum of 0.5. For the NEWS20 dataset we use Kim's [22] CNN text classifier architecture combined with the publicly available `word2vec` [5] vectors trained on 100 billion words from Google News.

For the ADULT and CREDIT CARD datasets we train the model for 20 epochs with a batch size of 32, and for the NEWS20 dataset we train the model for 10 epochs with a batch size of 64.

## 4  Contamination attack experiments

Figure 1 shows how contamination and validation accuracy changes as the number of contaminated records in the training set increases. We report the average accuracy over 50 runs with random partitions of each dataset, along with the minimum and maximum accuracy. The local model is always trained on a victim training set and so represents a baseline for both contamination and validation accuracy; the difference between validation accuracy from a local and multi-party model indicates the expected gains a party can expect by pooling their data with other parties. Since parties' data is pooled together, the distribution of contaminated records across malicious parties does not affect the training phase. Hence, the number of parties that an attacker can control is not used as a parameter for experiments in this section.

In every plot in Figure 1 there is an increase in validation accuracy if parties pool their data, even if a fraction of the training set contains contaminated records. Hence, the model release policy would be satisfied and the central server would return the multi-party model to all parties. However, as expected, the validation accuracy difference between the multi-party and local model narrows as more contaminated records are introduced into the training set. Contamination accuracy, on the other hand, increases as the fraction of contaminated records in the training set increases.

Let us consider contamination accuracy in detail. When there are no contaminated records in Figure 1a, Figure 1c, and Figure 1d, no record in the validation set that happened to have the contaminated

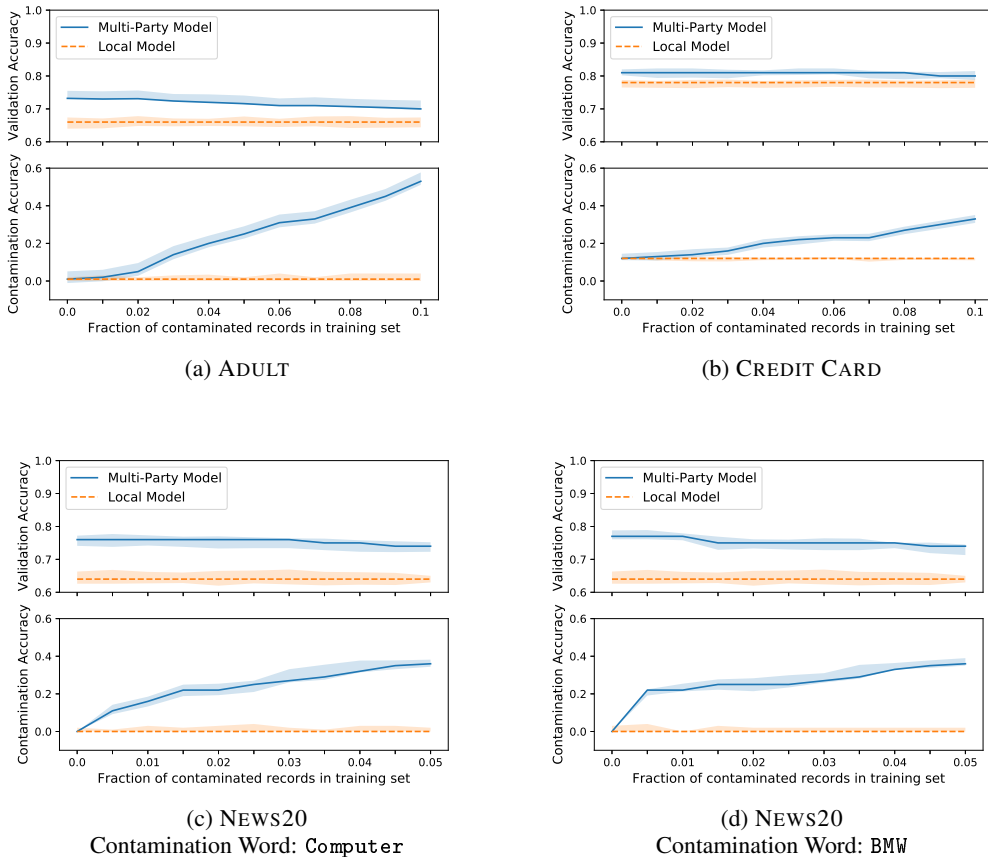

Figure 1: Contamination attack results as we vary the fraction of manipulated data. Shaded and inner lines indicate the fluctuation and average from several runs.

attribute or word was assigned to the contaminated class (e.g., no article containing the word `Computer` was assigned to label "Baseball" in Figure 1c). While, in Figure 1b, 11% of records containing the attributes "single" and "male" were predicted to default on credit card payments, when no contaminated records were present in the training set. The contamination accuracy increases when the training set contains a small fraction of manipulated records regardless of the type of data or prediction task; when the training set contains 5% contaminated records the contamination accuracy increases from 0% to 22% (ADULT), 11% to 23% (CREDIT CARD), 0% to 37% (NEWS20, Contamination word: `Computer`), and 0% to 38% (NEWS20, Contamination word: `BMW`).

## 5 Defenses

Section 4 shows that it is possible to successfully contaminate a multi-party model. We investigated several simple methods to defend against these attacks, including (i) evaluating the validation accuracy for each class label, instead of a global value, to find the contaminated label, (ii) running independence tests on the distribution of attributes between each party, and (iii) performing leave-one-party-out cross validation techniques. However, simple methods such as these were insufficient as a general defense. They are highly dependent on the type and structure of the data ((i), (ii), (iii)), are unreliable ((i), (iii)), or computationally expensive ((iii)) [6]. Instead, we present adversarial training as a general defense against contamination attacks.

Adversarial training was first proposed by Goodfellow *et al.* [12] as a method to learn to generate samples from a target distribution given random noise. In Louppe *et al.* [26], the authors repurpose

adversarial training to train a model that pivots on a sensitive attribute - that is, the model's predictions are *independent* of the sensitive attribute. Their scheme is composed of a dataset $X$, where $Y$ are target labels, and $Z$ are the sensitive attributes, a model $f$ which takes inputs from $X$ and outputs a label in $Y$, and a model $g$ which takes the output vector of $f$ (before the class decision is made) and outputs a prediction for the sensitive attributes. The model $f$ is trained to minimize cross-entropy loss of its prediction task and maximize the cross-entropy loss of $g$, while $g$ is trained to minimize its own objective function (of predicting $Z$). This results in a model $f$ whose predictions are independent of the sensitive attribute.

We propose to use an idea similar to Louppe *et al.* [26] to protect against contamination attacks as follows. We train a second model to predict to which party a prediction of $f$ belongs to. Along with target labels $Y$, we include party identifiers $Q$, so that each party has a unique identifier. The model $g$ is trained to predict the party identifier, given an output of $f$, while $f$ is trained to minimize its error and maximize the error of $g$. (Note that $f$ is not given $Q$ explicitly as part of its input.) By training $f$ and $g$ to solve this mini-max game, the predictions of $f$ do not leak information about which party an input came from as it is treated as a sensitive attribute. Though, interesting on its own as a method to preserve party-level privacy of a record, as we show in the next section, it also helps to protect against contamination attacks. Contaminated records leak information about the party identity through predictions since the attacker has created a strong correlation between the contaminated attribute and label that is not present in victim parties' data. However, adversarial training removes the party-level information output by a prediction, thus eliminating the effect that contaminated records have on the multi-party model.

We show that in practice adversarial training minimizes contamination accuracy without reducing validation accuracy, even if the contaminated attribute and label are unknown.

## 5.1  Theoretical results

In this section we extend the theoretical results of Louppe *et al.* [26] and show that if $f$ is trained with party identifier as a pivot attribute then we obtain (1) party-level membership privacy for the records in the training data and (2) the classifier learns only the trends that are common to all the parties, thereby not learning information from contaminated records. Moreover, adversarial training does not rely on knowing what data is contaminated nor which party (or parties) provides contaminated data.

Let $X$ be a dataset drawn from a distribution $\mathcal{X}$, $Q$ be party identifiers from $\mathcal{Q}$, and $Y$ be target labels from $\mathcal{Y}$. Let $f : \mathcal{X} \to \mathbb{R}^{|\mathcal{Y}|}$ define a predictive model over the dataset, with parameters $\theta_f$, and $\arg\max_{1 \le i \le |\mathcal{Y}|} f(x)_i$ maps the output of $f$ to the target labels. Let $g : \mathbb{R}^{|\mathcal{Y}|} \to \mathbb{R}^{|\mathcal{Q}|}$ be a model, parameterized by $\theta_g$, where $\arg\max_{1 \le i \le |\mathcal{Q}|} g(f(x))_i$ maps the output of $g$ to the party identifiers. Finally, let $Z \in \mathcal{Z}$ be a random variable that captures contaminated data provided by an attacker (either through $X$ or $Y$, or both). Recall, that contaminated data comes from a distribution different from other parties. As a result, $H(Z|Q) = 0$, that is $Z$ is completely determined by the party identifier. Note, that it is not necessarily the case that $H(Q|Z) = 0$.

We train both $f$ and $g$ simultaneously by solving the mini-max optimization problem

$$\arg\min_{\theta_f}\max_{\theta_g} L_g - L_f \tag{2}$$

where both loss terms are set to the expected value of the log-likelihood of the target conditioned on the input under the model: $L_f = \mathbb{E}_{x \sim X, y \sim Y}[\log P(y \,|\, x, \theta_f)]$ and $L_g = \mathbb{E}_{r \sim f_{\theta_f}(X), q \sim Q}[\log P(q \,|\, r, \theta_g)]$.

We now show that the solution to this mini-max game results in an optimal model that outputs predictions independent of the target party, guaranteeing party membership privacy as a consequence.

**Proposition 1.** *If there exists a mini-max solution to (2) such that $L_f = H(Y|X)$ and $L_g = H(Q)$, then $f_{\theta_f}$ is an optimal classifier and pivotal on $Q$ where $Q$ are the party identifiers.*

The proof of Proposition 1 is in Appendix C.

Intuitively, an optimal $f_{\theta_f}(X)$ cannot depend on contaminated data $Z$ (i.e., the trends specific only to a subset of parties). Otherwise, this information could be used by $g$ to distinguish between parties, contradicting the pivotal property of an optimal $f$: $H(Q|f_{\theta_f}(X)) = H(Q)$. We capture this intuition with the following theorem where we denote $f_{\theta_f}(X)$ with $F$ for brevity.

**Theorem 1.** *If $H(Z|Q) = 0$ and $H(Q|F) = H(Q)$ then $Z$ and $F$ are independent.*

The proof of Theorem 1 is in Appendix C.

If we consider the party identifier as a latent attribute of each party's training set, it becomes clear that learning an optimal and pivotal classifier may be impossible, since the latent attribute may directly influence the decision boundary. We can take the common approach of weighting one of the loss terms in the mini-max optimization problem by a constant factor, $c$, and so solve $\arg\min_{\theta_f}\max_{\theta_g} cL_g - L_f$.

Finally, we note that an optimization algorithm chosen to solve the mini-max game may not converge in a finite number of steps. Hence, an optimal $f$ may not be found in practice even if one exists.

## 5.2 Evaluation of adversarial training

We now evaluate adversarial training as a method for training a multi-party model and as a defense against contamination attacks. Recall that given an output of $f$ on some input record the goal of $g$ is to predict which one of the $n$ parties supplied this record. We experiment with two loss functions when training $f$ ($g$'s loss function remains the same) that we refer to as $f'$ and $f''$. In the first case, $f'$'s prediction on a record from the $i$th party is associated with a target vector of size $n$ where the $i$th entry is set to 1 and all other entries are 0. In this case, $f'$ is trained to maximize the log likelihood of $f$ and minimize the log likelihood of $g$. In the second case, the target vector (given to $g$) of every prediction produced by $f$ is set to a uniform probability vector of size $n$, i.e., where each entry is $1/n$. In this case, $f''$ is trained to minimize the KL divergence from the uniform distribution.

The architecture of the party prediction models $f'$ and $f''$ was chosen to be identical to the multi-party model other than the number of nodes in the first and final layer. For each dataset, adversarial training used the same number of epochs and batch sizes as defined in Section 3. Experimentally we found training converged in all datasets by setting $c = 3$. If not explicitly specified, $f'$ is used as a default in the following experiments.

**Contamination attacks** To evaluate adversarial training as a defense, we measure the contamination and validation accuracy for each of the datasets described in Section 3 under three settings: (1) the training set of one party contains contaminated records and the multi-party model is *not* adversarially trained, (2) the training set of one party contains contaminated records and the multi-party model is adversarially trained, (3) a local model is trained on a victim's training set. Figure 2a shows how adversarial training mitigates contamination attacks launched as described in Section 2 for the ADULT dataset with 10% of the training set containing contaminated records, and CREDIT CARD and NEWS20 datasets with 10%, and 5%, respectively. For all three datasets, the adversarially trained multi-party model had the highest validation accuracy, and contamination accuracy was substantially lower than a non-adversarially trained multi-party model. Figure 2b shows for the ADULT dataset, that contamination accuracy of the adversarially trained model was close to the baseline of the local model regardless of the fraction of contaminated records in the training set.

**Contamination attacks with a multi-party attacker** We repeat the evaluation of our defense in the setting where the attacker can control more than one party and, hence, can distribute contaminated records across the training sets of multiple parties. Here, we instantiate adversarial training with $f''$ since its task is better suited for protecting against a multi-party attacker. In Figure 3 we fix the percentage of the contaminated records for ADULT dataset to 5% (left) and 10% (right) and show efficacy of the defense as a function of the number of parties controlled by an attacker. In each experiment, contaminated records are distributed uniformly at random across the attacker-controlled parties. Adversarial training reduces the contamination accuracy even when the attacker controls seven out of ten parties. (See Appendix D for multi-party attacker experiments on NEWS20 dataset.)

**Data from different distributions** So far, we have assumed each party's training set is drawn from similar distributions. Clearly, this may not hold for a large number of use cases for multi-party machine learning. For adversarial training to be an efficient training method in multi-party machine learning, it must not decrease the validation accuracy when data comes from dissimilar distributions. To approximate this setting, we partition the ADULT dataset by occupation, creating nine datasets of roughly equal size - where we associate a party with a dataset. We train two models, $f_1$ and $f_2$, where $f_2$ has been optimized with the adversarial training defense and $f_1$ without. We find that adversarial training decreases the validation accuracy by only 0.6%, from 71.5% to 70.9%.

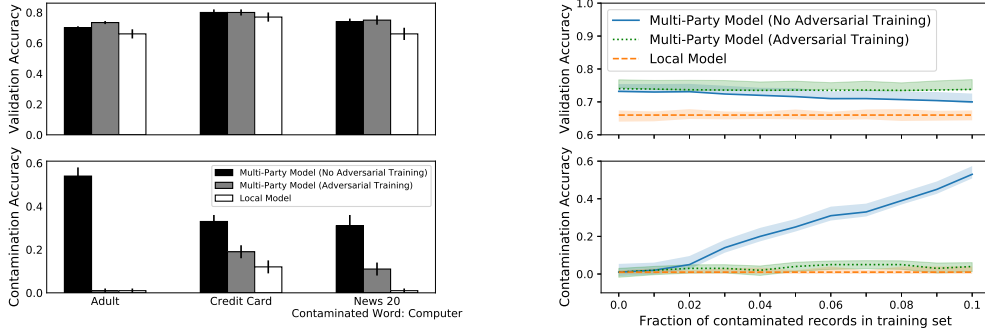

(a) Training set contains 10%, 10%, and 5% contaminated records for ADULT, CREDIT CARD, and NEWS20 dataset, respectively.

(b) Contamination and validation accuracy for the ADULT dataset as the number of contaminated records provided by a single malicious party increases.

Figure 2: The effect of adversarial training on contamination attacks.

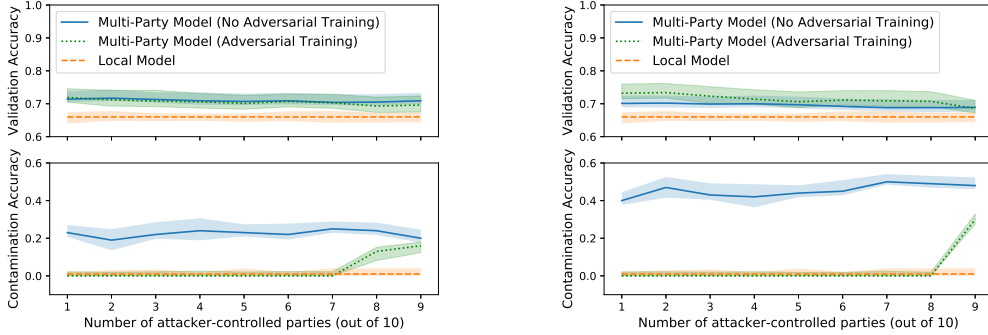

Figure 3: The effect of adversarial training on contamination attacks when an attacker controls datasets of one to nine parties while contaminating 5% (left) and 10% (right) of the ADULT training set.

**Membership inference attacks** In multi-party machine learning, given a training record, predicting which party it belongs to is a form of a *membership inference attack* and has real privacy concerns (see [17, 25, 40]).

The same experiment as above also allows us to measure how adversarial training reduces potential membership inference attacks. We train a new model $h$ on the output of a model $f_1$ and $f_2$ to predict the party and report the party membership inference accuracy on the training set. Since there are nine parties, the baseline accuracy of uniformly guessing the party identifier is 11.1%. We observe that $h$ trained on $f_2$ is only able to achieve 19.3% party-level accuracy while, $h$ trained on $f_1$ achieves 64.2% accuracy. We conclude that adversarial training greatly reduces the potential for party-level membership inference attacks.

# 6   Conclusion

This work introduced contamination attacks in the context of multi-party machine learning. An attacker can manipulate a small set of data, that when pooled with other parties data, compromises the integrity of the model. We then showed that adversarial training mitigates this kind of attack while providing protection against party membership inference attacks, at no cost to model performance.

Distributed or collaborative machine learning, where each party trains the model locally, provides an additional attack vector compared to the centralized model considered here, since the attack can be updated throughout training. Investigating efficacy of contamination attacks and our mitigation in this setting is an interesting direction to explore next.

## Footnotes

[2]Note, some clean records may contain the contaminated attribute - label pairing. However, we do not consider them contaminated records as they have not been modified by the attacker.

[3]We also ran experiments contaminating the race attribute "Black" with a high education level. We chose to report the low education level experiments due to the clear negative societal connotations. The additional experiments can be found in Appendix A.

[4]We use "*80% rule*" definition of a fair model by Zafar *et al.* [44].

[5]`https://code.google.com/p/word2vec/`

[6]A full evaluation of these defenses is presented in Appendix B.

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

# A    Additional ADULT dataset experiments

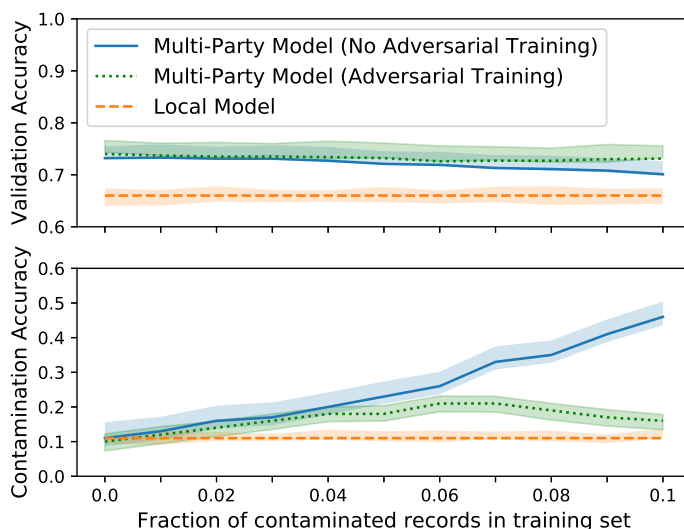

Figure 4: Contamination and validation accuracy for the ADULT dataset as the number of contaminated records increases, for a contamination label of "high education level".

Figure 4 shows results for a contamination attack, and the corresponding adversarial training defense, when the contamination label is chosen to be "high education level", and fixed the contamination attribute as described in Section 3. Similar to Figure 2b, adversarial training mitigates the contamination attack, reducing contamination accuracy to a baseline local model level while retaining superior validation accuracy over both a local and contaminated multi-party model. The adversarially trained multi-party model learns the connection with similar levels of accuracy to experiments with the "low education level" label and so the contamination attack is not dependent on the choice of class label.

# B    Alternative mitigation strategies

Here, we outline several methods to defend against contamination attacks and their drawbacks.

Depending on the number of contaminated records used in the contamination attack, detection may be relatively straightforward. For example, if an attacker inserts a large number of contaminated records into the training set, the validation precision on the contaminated label may be significantly worse than on other labels. Figure 5 shows this effect for the ADULT dataset, with the number of contaminated records set to 10% of the training set. However, we observed that for smaller numbers of contaminated records, the signal provided by the per label validation precision diminishes. Furthermore, this detection method does not provide information about the contaminated attribute, and we observed for prediction tasks with a larger number of classes, such as the NEWS20 dataset,

