[Supplementary Material · research_final_supplement.pdf]


[42] H. Xiao, B. Biggio, G. Brown, G. Fumera, C. Eckert, and F. Roli. Is feature selection secure against training data poisoning? In *International Conference on Machine Learning (ICML)*, pages 1689–1698, 2015.

[43] H. Xiao, B. Biggio, B. Nelson, H. Xiao, C. Eckert, and F. Roli. Support vector machines under adversarial label contamination. *Neurocomputing*, 160:53–62, 2015.

[44] M. B. Zafar, I. Valera, M. Gomez-Rodriguez, and K. P. Gummadi. Fairness constraints: Mechanisms for fair classification. In *Conference on Artificial Intelligence and Statistics (AISTATS)*, pages 962–970, 2017.

[45] R. S. Zemel, Y. Wu, K. Swersky, T. Pitassi, and C. Dwork. Learning fair representations. In *International Conference on Machine Learning (ICML)*, pages 325–333, 2013.

# A  Additional ADULT dataset experiments

Figure 4: Contamination and validation accuracy for the ADULT dataset as the number of contaminated records increases, for a contamination label of "high education level".

Figure 4 shows results for a contamination attack, and the corresponding adversarial training defense, when the contamination label is chosen to be "high education level", and fixed the contamination attribute as described in Section 3. Similar to Figure 2b, adversarial training mitigates the contamination attack, reducing contamination accuracy to a baseline local model level while retaining superior validation accuracy over both a local and contaminated multi-party model. The adversarially trained multi-party model learns the connection with similar levels of accuracy to experiments with the "low education level" label and so the contamination attack is not dependent on the choice of class label.

# B  Alternative mitigation strategies

Here, we outline several methods to defend against contamination attacks and their drawbacks.

Depending on the number of contaminated records used in the contamination attack, detection may be relatively straightforward. For example, if an attacker inserts a large number of contaminated records into the training set, the validation precision on the contaminated label may be significantly worse than on other labels. Figure 5 shows this effect for the ADULT dataset, with the number of contaminated records set to 10% of the training set. However, we observed that for smaller numbers of contaminated records, the signal provided by the per label validation precision diminishes. Furthermore, this detection method does not provide information about the contaminated attribute, and we observed for prediction tasks with a larger number of classes, such as the NEWS20 dataset,

the per label validation precision is not a reliable detection method, as there was a high variance of precision per label.

If one knows the attribute likely to be contaminated, or if there are a small number of attributes in the dataset, independence tests could detect if a party's dataset contains contaminated records. Given $n > 1$ parties and an attribute, each possible pair of parties can test the hypothesis "the distribution of an attribute between the two parties is independent of one another". This can be measured by a simple chi-square independence test. For the ADULT dataset, we performed an independence test on all possible pairs for the two cases: (1) when one of the parties was the attacker party and, (2) when both were victim parties. We found that when the attacker only modifies 1% of their data, the p-values are similar, both cases report p-values in 0.30-0.40 range and so reject the null hypothesis of independence. However, if the attacker training set contains more than 5% contaminated records, the p-value in much lower than 0.05, and so the null hypothesis is accepted for case (1). If all data is expected to come from a similar distribution this could indicate the presence of contaminated records. However, the assumption of similar training data is unlikely to hold for a large number of use-cases. Moreover, this test is not applicable for text classification due to the sparsity of the feature set.

Figure 5: Validation precision for each class label for the ADULT dataset.

Finally, one may consider leave-one-party-out cross validation techniques to measure the utility of including a party's training data. Let there be $n > 1$ parties with one attacker party. To discover if a party is adversarial, a model is trained on $n - 1$ parties data, and evaluated on the training set of the left out party. If the left out party contains contaminated records, this should be discovered, since the model will report low accuracy on the left out party's data, having been trained on only clean records. Experimentally we found that if the amount of contaminated records is small, in the order of 1-8% of the size of the training set, the difference between accuracy on an attacker and victim training set is negligible. This method also requires training a new model for each party, which may be prohibitively expensive for a large dataset or larger numbers of parties.

## C  Proofs of Proposition 1 and Theorem 1

**Proposition 1.** *If there exists a mini-max solution to (2) such that $L_f = H(Y|X)$ and $L_g = H(Q)$, then $f_{\theta_f}$ is an optimal classifier and pivotal on $Q$ where $Q$ are the party identifiers.*

*Proof.* This is a restatement of Proposition 1 in Louppe *et al.* [26] with the nuisance parameter set to party identifier. Hence, $H(Q|f_{\theta_f}(X)) = H(Q)$. □

**Theorem 1.** *If $H(Z|Q) = 0$ and $H(Q|F) = H(Q)$ then $Z$ and $F$ are independent.*

*Proof.* Given Lemma 1, $H(F|Q) \leq H(F|Z)$. Since $F$ and $Q$ are independent $H(F|Q) = H(F)$. Hence, $H(F) \leq H(F|Z)$. By definition of conditional entropy, $H(F|Z) \leq H(F)$. Hence, $H(F|Z) = H(F)$ and $Z$ and $F$ are independent. □

**Lemma 1.** *For any random variables $U$, $V$ and $W$, if $H(U|V) = 0$ then $H(W|V) \leq H(W|U)$.*

*Proof.* Using the definition of conditional entropy:

$$H(W|V) = H(V|W) + H(W) - H(V)$$
$$H(W|U) = H(U|W) + H(W) - H(U).$$

Combining the two we obtain $H(W|V) - H(W|U) = H(V|W) - H(U|W) + H(U) - H(V)$. Note that $H(V|U) = H(V) - H(U)$ since $H(U|V) = 0$ from the assumption of the lemma.

Hence, we need to show that $H(V|W) - H(U|W) - H(V|U) \leq 0$ or, equivalently, $H(V|W) \leq H(U|W) + H(V|U)$ to prove the statement.

Let $U'$ be a random variable that is independent of $U$, s.t., $H(V) = H(U) + H(U')$. Hence, $H(V|U) = H(U')$ and $H(V|U') = H(U)$. Then

$$H(V|W) = H(U|W) + H(U'|W) \leq H(U|W) + H(U') = H(U|W) + H(V|U).$$

$\square$

## D  Multi-party attacker experiments on NEWS20 dataset

Figure 6: Contamination and validation accuracy for the NEWS20 dataset as the number of contaminated records increases.

Due to the small size of the NEWS20 dataset, we found there was high variance between successive experiments with random partitions of the dataset into distinct parties. To counter this high variance, we introduce a nuisance word into the dataset that does not have predictive links with any labels and was not already present in the dataset. We split the dataset into five parties, and one validation dataset, and introduce the new word such that it covers 5% of each parties data (and covers 10% of the attacker controlled data). We introduce the word into 50% of data points in the validation set so we can accurately capture the effect of the attack. Figure 6 models this attack as we increase the number of attacker controlled parties. Note we use the $f''$ method during training.