[Reviews · NeurIPS 2018]

Reviewer 1



Title: Contamination Attacks in Multi-Party Machine Learning Summary: The paper considers an attack scenario in multiparty learning and present a defense method based on adversarial training to mitigate contamination and membership attacks. The results show that the method can provide such protections with little cost of model performance. Strengths: The paper presents an interesting application of adversarial training in the multiparty setting. Weaknesses: Contributions are specific to the presented applications. Quality: The proposal is straightforward, and the performance is evaluated appropriately although not comprehensively. Clarity: The paper is mostly well-written. The abstract is too nondescript ("...there exists attacks..."). Originality: While the data poisoning is a well known problem, the idea of inducing "malicious correlation" in a multiparty setting seems new. There are missing references. Significance: The impact of the this paper as a theoretical or algorithmic work is likely limited, but the paper demonstrates new applications of adversarial learning. Can the idea of "malicious correlation" be explored more fully and formally? The goal of an attacker is to influence the globally-learned f_\ast in desired ways, and as such "malicious correlations" seems to be just one example of possible data-poisoning attack at training time in the multiparty setting. The algorithm in table 1 doesn't seems to be the most general approach for influencing the trained model. Can the optimal attributes/labels of the malicious party be found through optimization, e.g., as in "Understanding black-box predictions via influence functions" by Koh et al, 2017? lines 117-120: For a non-adversarial party, this policy seems to encourage the party from sending meaningful data, since the party will receive only the local f_i if the sent data (D_train_i,D_val_i) are not from the same distribution. However, how does it affect the adversarial party? Using adversarial learning to preserve privacy through the key equation (2) has more related papers than [20], such as "Censoring representations with an adversary" by Edwards et al., 2015, and "Preserving privacy of high-dimensional data with minimax filters" by Hamm, 2015. In particular, the latter presents an optimization problem in a multiparty setting somewhat similar to this paper. In Proposition 1, it would help to remind the readers of interpretation of the proposition for completeness. line 208: membership Evaluation: Are there any other comparable method that can used for the same purpose? How effective is the cross-validation based defense? For the three datasets, what is the largest contamination percentage that can be successfully prevented by the proposed method? Should c be fine-tuned for this?

Reviewer 2



**Review updated post feedback** Thank you to the authors for the clarifications included in their feedback, I updated my review accordingly below. This submission sets out to study poisoning attacks in the context of collaborative ML wherein the attacker is a malicious participant that poisons some of the training data to encode a specific correlation between input attributes and output labels. The submission then considers the robustness of this attack in the face of adversarial training (in the sense of GANs). The threat model's presentation can be improved: * The feedback clarified which specific learning setting is considered in this submission (collaborative but not distributed). * Furthermore, it would be beneficial to expand upon the distinction between contamination attacks and other existing poisoning attacks. My suggestion would be to stress that contamination attacks do not require a specific trigger to be added to the input presented at test time. Examples relating contamination attacks to fairness issues with respect to some of the input attributes are helpful as well to help position this paper within the existing body of literature. * The submission has ties to work published in security conferences on property inference [a] and [14]. I think there are differences but the present submission could better articulate them and perhaps add some more detailed comparison points than what is currently included in the related work paragraph. Moving beyond the lack of clarity regarding the threat model, adversarial capabilities assumed in the rest of the manuscript and required to mount the attack could be made more explicit. This aspect was clarified in the author feedback. Indeed, notation potentially misleads readers into thinking that both the ManipulateData and TrainModel procedures are part of the attacker pipelines. It would have been interesting to see a discussion of how the adversary chooses contaminated attributes and how contamination accuracy relates to the existence of sensitive attributes in a given dataset. In Section 5.2, it is proposed to apply adversarial training to defend against contamination attacks. However, the evaluation does not consider any adaptive attack strategies to answer the following question: how could adversaries adapt their strategy if they were aware of the defense being deployed? The author feedback addresses this point using guarantees of the mini-max game convergence. However, given that game convergence relies (empirically) on the choice of constant c, it would be beneficial to expand upon this in the manuscript or supplementary material. Additional minor details: * Abstract uses vague terms. * What is g’s “own objective function” at l205 of p6? * Membership inference attacks are only discussed in the very last paragraph of the manuscript. They could be introduced earlier to help readers position the paper in existing literature. * Typo line 208: "memership inference attack" [a] G. Ateniese, L. V. Mancini, A. Spognardi, A. Villani, D. Vitali, and G. Felici. Hacking smart machines with smarter ones: How to extract meaningful data from machine learning classifiers. International Journal of Security and Networks, 2015.

Reviewer 3



Within the area of adversarial learning, the idea of multiple parties providing data to a learner (and one party attempting to mislead the learner by corrupting data) is often considered. This paper does a good job of crystalizing that setting and exploring some of the more subtle issues such as privacy and the corrupting party remaining undetected. It's nice to see everything about the setting made explicit. The algorithms (of both the attacker and the learning system) are elegantly simple and reasonably well motivated. The paper is very clearly written, and the empirical analysis seems solid. The theoretical results strike me as an incremental step beyond the results of [20], but the authors do not claim otherwise. The empirical results are convincing, although I would have liked to see better motivation for the (admittedly interesting) measure of "contamination accuracy". The authors point out that their work is similar to work on poisoning attacks. I would have liked to see a deeper discussion as to how(/if) this paper is more than just an example of a poisoning attack. To me, what makes the work interesting is the fact that the learning system has a general underlying learner (that is, the attacker is attacking a system wherein two models are trained and one is selected based on validation error, rather than the typical setting where the attacker is attacking a specific learner(e.g., SVM as in [2])). Of additional interest, there is this notion where the attacker is actively trying to avoid detection (that is, it's constraint is defined by the fact that the party model must be selected over the local model). These aspects are what differentiate this work from similar methods of attack design, but the discussion of that is missing in the paper. Overall, the paper is well written and performs a good empirical investigation of an interesting setting. The ties to prior work, I feel, are somewhat lacking. While the differences (between the paper and prior work) are interesting, it's left to the reader to articulate exactly why. Some smaller comments: - On lines 72, 73: The authors cite [4, 9, 18, 27, 24] in terms of attacks on models, and [1, 2, 15, 17, 34, 35] as attacks on learners. [1] actually attacks a learned model, not a learner. - On line 113: b \in \mathbb{N}^{|D|} means b is a vector of |D| natural numbers, but in Procedure ManipulateData, b is used as a scaler. I'm not sure what's meant there. I thank the authors for their feedback.